# Activation of Nuclear Factor Erythroid 2-Related Factor 2 Transcriptionally Upregulates Ectonucleotide Pyrophosphatase/Phosphodiesterase 1 Expression and Inhibits Ectopic Calcification in Mice

**DOI:** 10.3390/antiox13080896

**Published:** 2024-07-24

**Authors:** Tomomi Ida, Hiroyuki Kanzaki, Miho Shimoyama, Syunnosuke Tohyama, Misao Ishikawa, Yuta Katsumata, Chihiro Arai, Satoshi Wada, Shugo Manase, Hiroshi Tomonari

**Affiliations:** 1Department of Orthodontics, School of Dental Medicine, Tsurumi University, Yokohama 230-8501, Kanagawa, Japan; idatomomi.jelico@gmail.com (T.I.); toyama-s@tsurumi-u.ac.jp (S.T.); tomonari-h@tsurumi-u.ac.jp (H.T.); 2Department of Anatomy, School of Dental Medicine, Tsurumi University, Yokohama 230-8501, Kanagawa, Japan; ishikawa-misao@tsurumi-u.ac.jp; 3Department of Oral and Maxillofacial Surgery, Kanazawa Medical University, Kanazawa 920-1192, Ishikawa, Japan; wada-s@kanazawa-med.ac.jp

**Keywords:** ectopic calcification, Nrf2, pyrophosphate (PPi), osteoblastic differentiation, ENPP1, anti-oxidation, oxidative stress

## Abstract

Calcification plays a key role in biological processes, and breakdown of the regulatory mechanism results in a pathological state such as ectopic calcification. We hypothesized that ENPP1, the enzyme that produces the calcification inhibitor pyrophosphate, is transcriptionally regulated by Nrf2, and that Nrf2 activation augments ENPP1 expression to inhibit ectopic calcification. Cell culture experiments were performed using mouse osteoblastic cell line MC3T3-E1. Nrf2 was activated by 5-aminolevulinic acid and sodium ferrous citrate. Nrf2 overexpression was induced by the transient transfection of an Nrf2 expression plasmid. ENPP1 expression was monitored by real-time RT-PCR. Because the promoter region of ENPP1 contains several Nrf2-binding sites, chromatin immunoprecipitation using an anti-Nrf2 antibody followed by real-time PCR (ChIP-qPCR) was performed. The relationship between Nrf2 activation and osteoblastic differentiation was examined by alkaline phosphatase (ALP) and Alizarin red staining. We used mice with a hypomorphic mutation in ENPP1 (ttw mice) to analyze whether Nrf2 activation inhibits ectopic calcification. Nrf2 and Nrf2 overexpression augmented ENPP1 expression and inhibited osteoblastic differentiation, as indicated by ALP expression and calcium deposits. ChIP-qPCR showed that some putative Nrf2-binding sites in the ENPP1 promoter region were bound by Nrf2. Nrf2 activation inhibited ectopic calcification in mice. ENPP1 gene expression was transcriptionally regulated by Nrf2, and Nrf2 activation augmented ENPP1 expression, leading to the attenuation of osteoblastic differentiation and ectopic calcification in vitro and in vivo. Nrf2 activation has a therapeutic potential for preventing ectopic calcification.

## 1. Introduction

Calcification is important in bone formation [1] and tooth development [2], but improperly controlled biomineralization can lead to pathological conditions such as ectopic calcification, which is the deposition of calcium salts in tissues and organs that induces organ dysfunction [3]. Therefore, ectopic calcification is associated with a number of diseases such as ligament ossification [4] and vascular calcification [5], and these disorders can be life-threatening [6].

Although the causes and mechanisms of ectopic calcification are unclear, several factors involved in calcification in vivo have been identified. Calcification occurs primarily by the reaction of phosphate (Pi) and calcium ions [7]. The concentrations of phosphate and calcium ions are strictly regulated, and any imbalance can disrupt cellular function [8]. That is, calcification is regulated by the enzymes involved in the metabolism of Pi and pyrophosphate (PPi) [9]. Pi is important in intracellular energy metabolism and signal transduction [10], and excess Pi reacts with calcium ions extracellularly, causing calcification [11]. By contrast, PPi, which is less reactive with calcium ions than phosphoric acid, prevents calcification [9]. Ectonucleotide pyrophosphatase/phosphodiesterase 1 (ENPP1) catalyzes the conversion of Pi to PPi [12], and progressive ankylosis gene product (ANK) catalyzes the reverse reaction [13]. Alkaline phosphatase (ALP) hydrolyzes phosphoric acid to produce inorganic Pi, which promotes calcification [14].

Genetic mutations in ENPP1, ANK, and ALP influence the development and progression of ectopic calcification; ENPP1 mutations inhibit the production of excess PPi and promote the reaction of Pi with calcium ions [12]. This causes ectopic calcification of the joints and skin. The ANK mutation conversely causes excessive PPi production and inhibits the reaction between Pi and calcium ions, resulting in the ectopic calcification of bone and cartilage [15]. Loss-of-function mutations in ALP cause hypophosphatasia, which attenuates normal calcification [16]. Therefore, ENPP1, ANK, and ALP are important in calcification in vivo, and their mutations affect the development and progression of ectopic calcification.

In ENPP1 deficiency, the PPi level in serum is low, resulting in a variety of phenotypes [17]. These include systemic arterial calcification (GACI), which causes the extensive calcification of large- and medium-sized arteries in early life [12] and autosomal recessive hypophosphatemic rickets type 2 (ARHR2), which causes acquired hypophosphatemic rickets [18]. ENPP1 mutations also cause pseudoxanthoma (PXE), a cutaneous systemic disease characterized by the ectopic calcification of elastic fibers [17]. These diseases are thought to share a molecular mechanism—PPi deficiency caused by reduced ENPP1 activity. Therefore, supplementary therapy for PPi could prevent ectopic calcification [19]. Furthermore, a recombinant ENPP1-Fc fusion protein [20] or recombinant human ENPP1 protein [21] could prevent vascular calcification. Therefore, ENPP1 augmentation may prevent ectopic calcification.

Distal-less homeobox 3 (DLX3) and other factors are reported to control ENPP1 expression [22,23,24,25,26,27,28,29,30]. In addition, oxidative stress downregulates ENPP1 expression [31]. We reported a relationship between osteoclastogenesis and anti-oxidation via the transcription factor, nuclear factor erythroid 2-related factor 2 (Nrf2) [32,33,34], which suggests a link between Nrf2 and ENPP1 expression. We hypothesized that ENPP1 is transcriptionally regulated by Nrf2, and that Nrf2 activation augments ENPP1 expression to inhibit ectopic calcification. To evaluate this hypothesis, we performed in vivo and in vitro assays.

## 2. Materials and Methods

### 2.1. Reagents

Beta glycerophosphate (*β*-GP), ascorbic acid (AA), recombinant bone morphogenetic protein 2 (BMP-2), 5-aminolevulinic acid (ALA), and Alizarin red were purchased from Fujifilm Wako Pure Chemical Corporation (Tokyo, Japan). Sodium ferrous citrate (SFC) was provided by Eisai Food and Chemical (Tokyo, Japan).

### 2.2. Cells

MC3T3-E1 mouse calvaria-derived cells were obtained from Riken BioResource Research Center (Tsukuba, Japan).

### 2.3. Expression Plasmid

The Nrf2 (accession number: NM_006164.5) expression plasmid (pcDNA3-EGFP-C4-Nrf2) [35] was from Addgene (Watertown, MA, USA).

### 2.4. Cell Culture

MC3T3-E1 cells were cultured in α-modified Eagle’s medium (α-MEM; Wako Pure Chemical, Osaka, Japan) that contained 10% fetal bovine serum (Thermo Scientific, South Logan, UT, USA) supplemented with antibiotics (100 U/mL penicillin and 100 μg/mL streptomycin) at 37 °C in a 5% CO_2_ incubator.

### 2.5. Calcification-Induction Assay

Cells were cultured in control or calcification medium (control medium supplemented with 0.01 μM dexamethasone and 50 μg/mL ascorbic acid) with or without Nrf2 activators (20 μM ALA and 10 μM SFC).

### 2.6. Gene Transfection

Nrf2 expression plasmids were transfected using X-tremeGeneHD DNA transfection reagent (Roche, South San Francisco, CA, USA) according to the manufacturer’s instructions. Briefly, cells were seeded onto 24-well plates at a density of 6.0 × 10^4^ cells/well and transfected with Nrf2 expression plasmid (2 μg/well) using X-tremeGeneHD DNA transfection reagent. Transfection medium was exchanged after 6 h, and cells were subsequently cultured for 3 days.

### 2.7. Real-Time RT-PCR

RNA was extracted from MC3T3-E1 cells using the NucleoSpin^®^ RNA (Macherey-Nagel, Düren, Germany) with on-column genomic DNA digestion according to the manufacturer’s instructions. After the measurement of the RNA concentration, RNA (500 ng each) was reverse-transcribed with iScript cDNASupermix (Bio-Rad Laboratories, Hercules, CA, USA), and cDNA was diluted (10×) with Tris-EDTA buffer. Real-time RT-PCR was performed using SsoFast EvaGreen-Supermix (Bio-Rad Laboratories). Primer sequences from PrimerBank (https://pga.mgh.harvard.edu/primerbank/) accessed on 17 May 2017 were mouse ribosomal protein S18 (Rps18, accession number: M76763), (F) 5′-AGT TCC AGC ACA TTT TGC GAG-3′ and (R) 5′-TCA TCC TCC GTG AGT TTC TCC A-3′ and mouse ENPP1 (accession number: M12552), (F) 5′-CGC CAC CGA GAC TAA A-3′ and (R) 5′-TCA TAG CGT CCG TCA T-3′. Fold-changes in expression were calculated by the ΔΔCt method using Rps18 as the reference gene.

### 2.8. Immunofluorescence Analysis of ENPP1

Cells were seeded onto coverslips (Matsunami Glass Ind., Ltd., Osaka, Japan) in six-well plates and cultured as described above for 5 days. Cultured cells were fixed with 4% paraformaldehyde in phosphate-buffered saline (PBS) for 1 h and washed three times. Subsequently, the cells were blocked with 10% bovine serum albumin in PBS for 1 h at room temperature. A primary antibody against ENPP1 (Bioss Inc., Woburn, MA, USA), followed by an Alexa Fluor 488-labeled secondary antibody (Abcam, Cambridge, UK), were added for 1 h at room temperature. Cells were stained with 4′,6-diamidino-2-phenylindole (DAPI) to visualize nuclei. Finally, coverslips were mounted, and slides were imaged using a BZ-9000 fluorescence microscope (Keyence, Osaka, Japan).

### 2.9. Measurement of PPi Concentration in Culture Supernatant

The PPi concentration in the culture supernatant was measured using the Pyrophosphate Assay Kit II (Fluorometric) (Abcam) according to the manufacturer’s instructions. Samples were diluted 20-fold to be within the detection range. To eliminate the influence of possible differences in numbers of cells, the protein concentration was measured using the Pierce Bicinconic Acid Assay (BCA) Protein Assay Kit (Thermo Scientific) and normalized to the concentration of the protein in the well.

### 2.10. Osteoblastic Differentiation Assay

Cells were stimulated with *β*-GP (10 mM) and AA (37.5 μg/mL) to induce osteoblastic differentiation. In some conditions, cells were treated with ALA (20 μM) and SFC (10 μM). Cells were subjected to RNA extraction on day 5 for the analysis of gene expression or stained for the osteoblastic differentiation marker ALP on day 10 and with Alizarin red on day 28.

### 2.11. In Silico Analysis of Promoter-Binding Regions

Putative Nrf2-binding sites (antioxidant responsive elements [AREs]) in the promoter region of ENPP1 (accession number: AC099695.15) were identified using PROMO (https://alggen.lsi.upc.es/) [36] and TFBIND (https://tfbind.hgc.jp/) [37] accessed on 29 May 2017. The common core sequence of ARE is the RTGACnnnGC motif [38].

### 2.12. Chromatin Immunoprecipitation (ChIP)

Cells were plated in a 100 mm dish, and on the next day, the medium was exchanged for fresh medium with or without ALA and SFC. On day 3, cells were subjected to ChIP and chromatin samples were prepared using the Pierce ChIP Kit (Thermo Scientific, Rockford, IL, USA) according to the manufacturer’s instructions. Immunoprecipitation was performed using normal rabbit IgG as the negative control and an anti-Nrf2 antibody (Cell Signaling Technology, Inc., Danvers, MA, USA) for samples. Precipitated chromatin samples were de-crosslinked with high-salt proteinase K solution at 65 °C for 2 h. DNA samples were prepared using spin columns and used for real-time PCR (ChIP-qPCR) using SsoFast EvaGreen-Supermix (Bio-Rad Laboratories). The ChIP-qPCR primers were sequence-1: (F) 5′-CTA CGG TTT AAT CCT ACT CTA GCC-3′ and (R) 5′-TAA AAC AAT GTG AAC ACT TTC AAT-3′, sequence-2: (F) 5′-GTA TCT GCA CAC ATG CAT GAA AGT-3′ and (R) 5′-TGC TAA GTT CAA GGC TAA TAA GAA-3′, sequence-3: (F) 5′-TGC CCT TAG AAT CTC ATG CTT GCG-3′ and (R) 5′-TGA AGT TTA ACG GGT GTA AAT TGG-3′, sequence-4: (F) 5′-CTT AGT CAC TGT GTT CAG TCA CAT-3′ and (R) 5′-GAT TCT TCA GCT TTA CAC TCC TTC-3′, sequence-5: (F) 5′-TGA AAC AAA TCC CAT CAA GAA TTA A-3′ and (R) 5′-GGT GTC TGT GTC TTG CTT CTC CGA-3′, sequence-6: (F) 5′-AGT TAA AGA CAC GCC CAC GTC AGG-3′ and (R) 5′-TCC GGC CCG AGC CGC TCA ACG CCC-3′. Results were expressed as % input.

### 2.13. ALP and Alizarin Red Staining

Cells were fixed with 4% paraformaldehyde in PBS and stained for ALP using the TRAP/ALP Stain Kit (Fujifilm Wako Pure Chemical Corporation) according to the manufacturer’s instructions. Cells were fixed with methanol and stained with Alizarin red for 15 min. The cells were washed with milli-Q water and imaged using a BZ-9000 microscope (Keyence, Osaka, Japan). The percentages of ALP- and Alizarin red-stained areas per field were calculated from five photographs for each culture condition using ImageJ software version 1.52a (National Institutes of Health, Bethesda, MD, USA).

### 2.14. Animal Experiment

The experimental protocols were approved by the Institutional Animal Care and Use Committee of Tsurumi University (approval number 29A057). Animals were treated ethically, and animal experiments were performed in compliance with the Regulations for Animal Experiments and Related Activities at Tsurumi University. Five-week-old male TWY/Jic (tiptoe walking (ttw)/ttw) mice (*n* = 10) were purchased from CLEA Japan, Inc. (Tokyo, Japan). Throughout the study, the experimental subjects were maintained in a controlled environment with a 12 h light/dark cycle and unrestricted access to nourishment and purified water. The research facility is equipped to regulate temperature, humidity, air circulation, and illumination optimally. Moreover, its robust infrastructure is designed to prevent animal escape while mitigating potential environmental disturbances such as odors, noise, and waste. After 1 week of acclimation, mice were divided into the control and Nrf2 activation groups (n = 5 each). The control group received an intraperitoneal injection of PBS, and the Nrf2 activation group received an intraperitoneal injection of ALA (100 μg/g BW) and SFC (157 μg/g BW). The injections were performed three times per week for 8 weeks. Food intake and body weight changes were monitored throughout the experimental period. Mice were euthanized at 14 weeks of age, and tissue specimens were fixed with 4% paraformaldehyde in PBS overnight.

### 2.15. microCT Analysis of Animal Samples

Specimens were scanned using an X-ray microtomography (microCT) system (inspeXio SMX-225CT; Shimadzu Corp., Kyoto, Japan) with the following settings: tube voltage, 140 kV; tube current, 70 mA; and slice thickness, 0.132 mm. After the reconstitution and generation of DICOM files, three-dimensional volume renderings and multi-planar reconstructions were performed using RadiAnt Dicom Viewer (Medixant, Poznań, Poland) using the same contrast and whiteness. The signal intensity of ligament tissue between the second (C2) and third cervical vertebrae (C3) was measured using ImageJ software version 1.52a. Briefly, the dorsal proximal part of the vertebral body of C2 and C3 in the images of the cervical vertebrae in the sagittal plane was set as the starting and ending point for the signal intensity analysis. The mean signal intensity of the ligament area between C2 and C3 was measured.

### 2.16. Statistical Analysis

Data are means ± standard deviation (SD) and scatterplots with a full dataset. Mean values were compared between two groups using a *t*-test. Multiple comparisons were performed using Tukey’s test. A value of *p* < 0.05 was considered indicative of statistical significance.

## 3. Results

### 3.1. Nrf2 Activation by Small Molecules Augmented ENPP1 Expression

Nrf2 activation by small molecules significantly augmented ENPP1 expression (Figure 1a). Interestingly, calcification-induction medium attenuated ENPP1 expression. By contrast, Nrf2 activation by small molecules in calcification-inducing medium overcame the attenuation and augmented ENPP1 expression. Immunofluorescence analysis revealed that Nrf2 activation augmented ENPP1 expression (Figure 1b). There was a statistically significant difference in immune-positive cells between the groups, and Nrf2 activation samples exhibited extensively higher percentage of positive area (Figure 1c). These results suggest that Nrf2 activation transcriptionally increases ENPP1 expression.

### 3.2. Exogenous Nrf2 Overexpression Augmented ENPP1 Expression

To confirm that Nrf2 augments ENPP1 expression, we overexpressed Nrf2 and examined ENPP1 expression. Real-time RT-PCR showed that exogenous Nrf2 overexpression augmented ENPP1 expression (Figure 1d).

### 3.3. Nrf2 Activation Increased the Extracellular PPi Concentration

We evaluated the effect of upregulated ENPP1 on extracellular PPi (Figure 1e). Nrf2 activation significantly increased the extracellular PPi concentration. Therefore, Nrf2 activation transcriptionally increases ENPP1 expression, leading to an increased concentration of extracellular PPi, an inhibitor of calcification.

### 3.4. Transcriptional Regulation of ENPP1 Expression by Nrf2

We performed a bioinformatics analysis to evaluate whether Nrf2 directly regulates the transcription of ENPP1. The promoter region of ENPP1 contains several putative Nrf2-binding sites (Figure 2). Therefore, we constructed PCR primer sets and performed ChIP-PCR to identify sites that participate in Nrf2-mediated ENPP1 transcriptional upregulation. The signals of analyzed sequences 1, 2, 3, and 4 were augmented in Nrf2 antibody-precipitated samples (Figure 3). The signal of sequences 5 was similar under all culture conditions. The signal of sequence 6 was attenuated by Nrf2 activation compared to the control. These results suggest that ENPP1 is transcriptionally regulated by Nrf2, in which sequences 1 to 4 are implicated.

### 3.5. Nrf2 Activation Inhibited Osteoblastic Differentiation and Calcification

ALP staining showed that calcification medium and BMP-2 induced ALP expression in vitro (Figure 4). Nrf2 activation attenuated ALP expression in the presence of BMP-2 stimulation (Figure 4a). Quantification of staining intensity showed that Nrf2 activation significantly reduced ALP expression (Figure 4b).

Alizarin red staining showed that calcification medium induced calcification in vitro (Figure 5a), an effect attenuated by Nrf2 activation. The quantification of staining intensity showed that Nrf2 activation significantly reduced calcification, and there was no significant difference between the control and Nrf2 activation conditions in calcification medium (Figure 5b).

### 3.6. Nrf2 Activation Blocked Ectopic Calcification in a Mouse Model of Ossification of the Posterior Longitudinal Ligament of the Spine (OPLL)

Finally, we examined whether Nrf2 activation blocked ectopic calcification in experimental animals. OPLL mice exhibited ectopic calcification in the posterior longitudinal ligament of the spine (Figure 6a,c). By contrast, an Nrf2 activator blocked ectopic calcification (Figure 6b,d). Quantification showed a significant difference in calcification between the spine and higher signal intensity in the control group than in the Nrf2 activation group (Figure 6e). Therefore, Nrf2 activation inhibited ectopic calcification by upregulating ENPP1.

## 4. Discussion

Here, we report that ENPP1 is transcriptionally regulated by Nrf2, and that Nrf2 activation augments ENPP1 expression to inhibit ectopic calcification in vitro and in vivo. ENPP1 generates extracellular inorganic PPi [39], an inhibitor of hydroxyapatite crystal deposition and growth [40]. Therefore, our results suggest that Nrf2 activation has prophylactic potential for ectopic calcification.

Although Nrf2-mediated ENPP1 augmentation blocked ectopic calcification, we did not measure extracellular PPi. Therefore, confirmation of the increased extracellular PPi concentration via Nrf2-mediated ENPP1 augmentation is needed. In addition, loss-of-function experiments for ENPP1 during Nrf2-mediated ENPP1 augmentation were not performed. Further studies are needed to examine whether a neutralizing antibody for ENPP1 during Nrf2-mediated ENPP1 augmentation attenuates Nrf2-mediated suppression of ectopic calcification.

Regulation of ectopic calcification involves exogenous Pi supplementation, which promotes osteoblastic differentiation of stem and progenitor cells [41]. By contrast, PPi inhibits calcification [42]. This Pi/PPi balance is regulated by several functional proteins. Alkaline phosphatase degrades PPi to Pi, thereby inducing calcification [43]. ENPP1 on the plasma membrane hydrolyzes ATP to produce AMP and PPi [20]. Mutation in ENPP1 resulted in a reduced PPi concentration, which attenuated local tissue calcification in mice [20]. Therefore, the augmentation of PPi via ENPP1 induction has potential as a therapeutic target for the prevention of ectopic calcification. Indeed, the administration of a recombinant ENPP1-Fc fusion protein [44] or recombinant human ENPP1 protein [21] prevented vascular calcification. In this research, ENPP1 augmentation by Nrf2 activation suppressed ectopic calcification in ENPP1 mutant mice, which is consistent with the regulatory mechanism of the Pi/PPi balance.

ENPP1 is implicated in the regulation of the extracellular Pi/PPi balance, so there have been attempts to control ectopic calcification using ENPP1. Nitschke et al. reported that recombinant ENPP-Fc protein prevented generalized arterial calcification of infancy [45]. In ENPP1-deficient mice, recombinant ENPP protein prevented nephrocalcinosis [46] and Achilles tendon calcification [47]. Recombinant ENPP1 also prevented ectopic calcification in a mouse model of heritable multisystem ectopic calcification disorder [48]. Therefore, the augmentation of ENPP1 via the administration of recombinant ENPP1 can prevent ectopic calcification in vivo. In this study, we showed that the augmentation of ENPP1 by transcriptionally upregulating ENPP1 expression prevented ectopic calcification.

The transcription factor DLX3 upregulates ENPP1 expression [22], as do fibroblast growth factors (FGFs) [23], heat shock protein 70 [24], TGF-β [25], osterix [28], and dentin matrix protein 1 [30]. By contrast, Spexin downregulates ENPP1 [49]. However, there is, to our knowledge, no report of the relationship between ENPP1 expression and Nrf2. Interestingly, oxidative stress reportedly decreases ENPP1 expression [31], suggesting a positive interaction between ENPP1 expression and Nrf2 [50]. A comprehensive review of the literature reveals intriguing insights into the complex relationship between oxidative stress and Nrf2 activation across various cell types. Notably, several studies have documented that oxidative stress-induced Nrf2 activation exhibits a transient nature in specific cellular contexts. For instance, a study published in Glia [51] demonstrated this transient activation pattern in astrocytes, while another investigation focusing on intervertebral disk cells, published in the *European Cells and Materials* journal [52], reported similar findings.

Further evidence supporting this concept comes from a study on bovine mammary epithelial cells [53]. This research utilized a sophisticated approach to measuring transcription activity by assessing the fluorescence activity of the antioxidant response element. The results unequivocally demonstrated that cells subjected to oxidative stress conditions exhibited a significant attenuation of transcription activity compared to the control groups. This observation provides crucial insights into the cellular response mechanisms under oxidative stress conditions. Synthesizing these findings, we can postulate that while oxidative stress may initially trigger a transient upregulation of Nrf2 activation, this does not necessarily translate into sustained Nrf2-mediated transcriptional responses. This apparent discrepancy between initial activation and subsequent transcriptional output suggests a more nuanced and complex regulatory mechanism at play. It is plausible that cells possess intricate feedback loops or compensatory mechanisms that modulate the Nrf2 pathway’s activity over time, potentially as a means of maintaining cellular homeostasis or preventing overactivation of the stress response pathways. These observations have profound implications for our understanding of cellular stress responses and the role of Nrf2 in particular. They challenge the simplistic view of a linear relationship between oxidative stress, Nrf2 activation, and downstream transcriptional effects. Instead, they point towards a more sophisticated, temporally regulated system that may involve multiple layers of control.

As an interpretation of the results of ChIP-qPCR, there are several possible reasons for the differences in signal intensity among the analyzed sequences. First, each analyzed sequence contains several putative Nrf2-binding sites (sequence-1: 2 sites, sequence-2: 2 sites, sequence-3: 2 sites, sequence-4: 3 sites, sequence-5: 4 sites, sequence-6: 1 site). The difference in this number of the amount of binding sites that are contained might have an effect on the results. Second, each putative Nrf2-binding site is different in the sequence, and these variations in sequence may potentially exert a substantial impact on the binding affinity of Nrf2 [38].

The transcriptional regulation of ENPP1 by Nrf2 likely involves the well-established mechanism of Nrf2 binding to Antioxidant Response Elements (AREs). Under normal conditions, Nrf2 is sequestered in the cytoplasm by Keap1, which facilitates its ubiquitination and subsequent proteasomal degradation [54]. Upon oxidative stress or electrophilic stimuli, Nrf2 escapes Keap1-mediated degradation, translocates to the nucleus, and binds to AREs to activate target gene expression [55]. Interestingly, Bach1 acts as a competitive inhibitor of Nrf2, binding to the same ARE sequences and repressing gene expression [56]. The interplay between Nrf2, Keap1, and Bach1 forms a complex regulatory network that fine-tunes the expression of antioxidant genes, including ENPP1, in response to cellular stress conditions. Furthermore, the possible transactivation of Nrf2 by vitamin D was recently reported [57], though, classically, vitamin D is reported to induce ectopic calcification [58]. These inconsistencies should be explored in the future.

## 5. Conclusions

Our research demonstrates that ENPP1 is subject to transcriptional regulation by Nrf2. Furthermore, we have established that the activation of Nrf2 leads to an upregulation of ENPP1 expression, which in turn inhibits ectopic calcification both in vitro and in vivo. To illustrate our findings, we have provided a graphical abstract that elucidates the proposed mechanism by which Nrf2 activation prevents ectopic calcification. This mechanism involves the Nrf2-mediated enhancement of ENPP1 expression, which subsequently promotes the conversion of inorganic phosphate (Pi) to pyrophosphate (PPi). PPi serves as a potent inhibitor of the calcification process.

The implications of our study are significant, as they suggest a novel approach to combating ectopic calcification. Specifically, our results indicate that the activation of Nrf2 holds considerable prophylactic potential in preventing abnormal tissue calcification. This finding opens up new avenues for therapeutic interventions targeting ectopic calcification, a pathological process implicated in various cardiovascular and metabolic disorders.

Our research not only advances our understanding of the molecular mechanisms underlying ectopic calcification but also highlights the importance of the Nrf2–ENPP1 axis in maintaining tissue homeostasis. These insights may pave the way for the development of innovative strategies to prevent or mitigate ectopic calcification, potentially improving outcomes for patients affected by this pathological process.

## Figures and Tables

**Figure 1 antioxidants-13-00896-f001:**
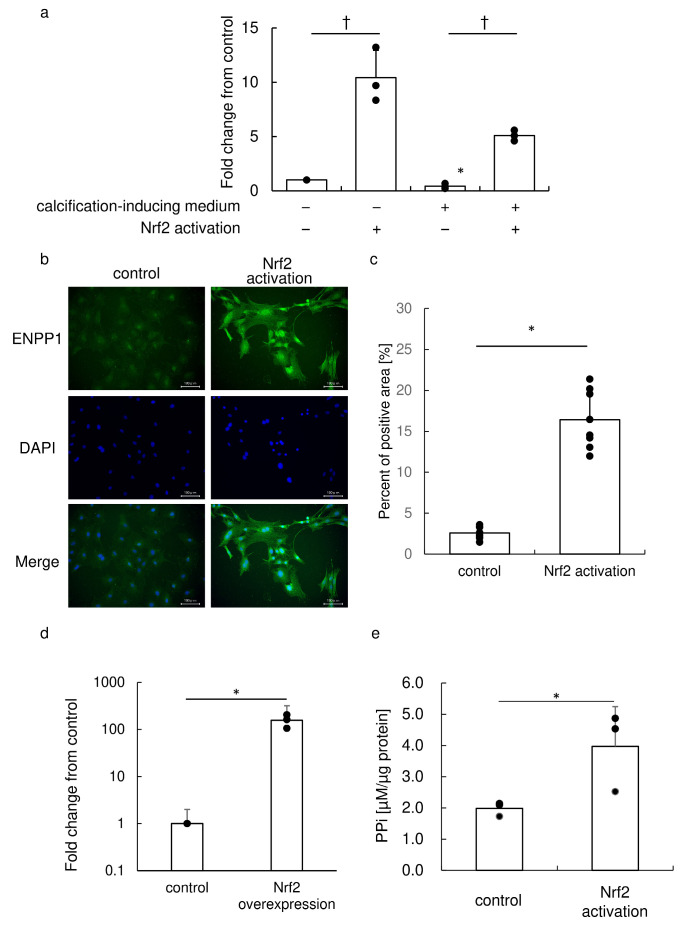
Nrf2 activation augments ENPP1 expression. (**a**) Real-time RT-PCR analysis of ENPP1 on day 5. Gene expression was calibrated using the Rps18 housekeeping gene, and values are fold-changes compared to the control. Data are representative of three independent experiments performed in triplicate. * *p* < 0.05 vs. control, † *p* < 0.05 between samples. (b) Immunofluorescence analysis of ENPP1 on day 5. Cells cultured in medium with or without Nrf2 activator were subjected to immunofluorescence staining with an anti-ENPP1 primary antibody and fluorophore-labeled secondary antibody. Photographs were obtained using an exposure of 1.3 s for ENPP1 and 1/80 s for DAPI. (c) Signal intensity of ENPP1 analyzed using ImageJ (N = 8). Data are means ± SD and scatterplots with full dataset. * *p* < 0.05. (d) The ENPP1 mRNA level was augmented by Nrf2 overexpression. Gene expression 3 days after gene transfection was calibrated using the Rps18 housekeeping gene, and values are fold-changes compared to the control. Data are representative of three independent experiments performed in triplicate. Data are means ± SD and scatterplots with full dataset. * *p* < 0.05 vs. control. (e) Nrf2 activation augmented extracellular PPi. Concentration of extracellular PPi normalized to the protein concentration is indicated. Data are means ± SD and scatterplots with full dataset. (N = 3). * *p* < 0.05.

**Figure 2 antioxidants-13-00896-f002:**
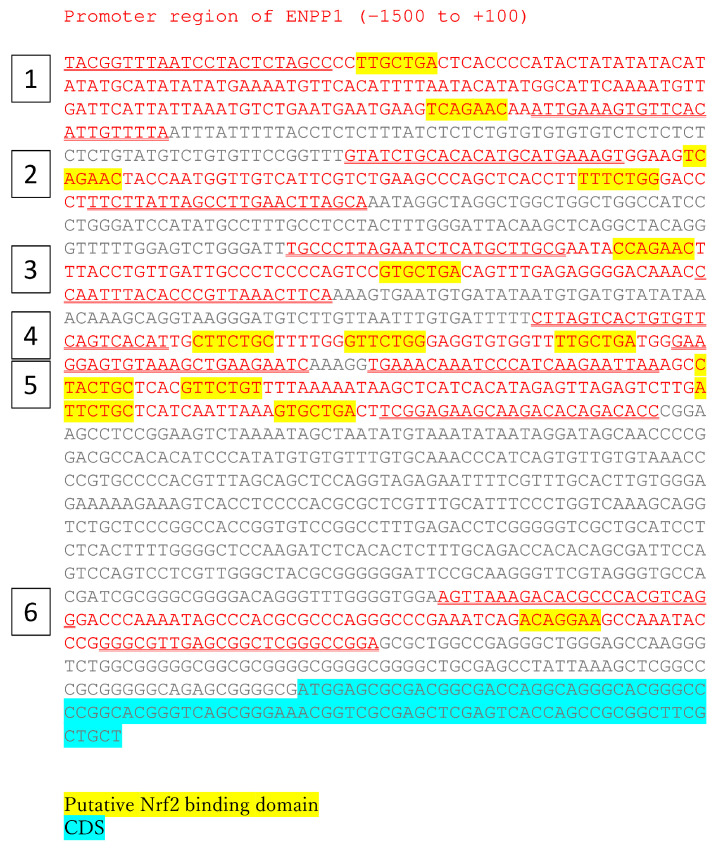
In silico analysis of putative Nrf2-binding sites in the promoter region of ENPP1. The sequence of the ENPP1 promoter region (1500 bp upstream to 100 bp downstream of the start codon) was examined for putative Nrf2-binding sites using web-based software as described in the materials and methods section. Yellow highlighting indicates putative Nrf2-binding sites. Analyzed sequences are in red and numbered at left. The ChIP-qPCR primers are underlined. CDS, coding sequence (cyan).

**Figure 3 antioxidants-13-00896-f003:**
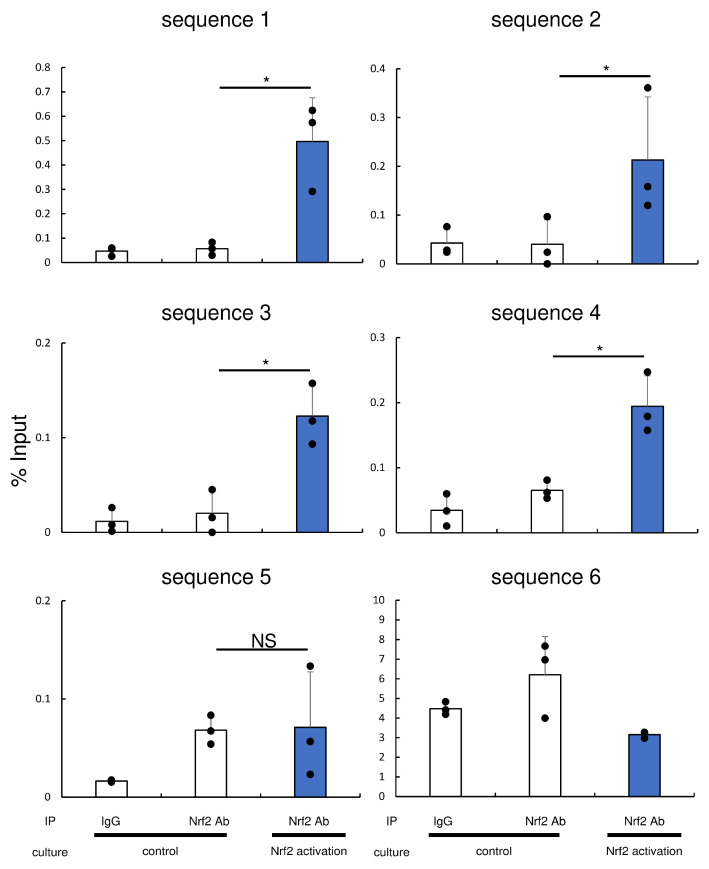
ChIP-qPCR analysis of the promoter region of ENPP1. Six sequences containing several putative Nrf2-binding sites shown in Figure 2 were analyzed by ChIP-qPCR. Cycle threshold (Ct) of percentage input DNA is shown. Samples immunoprecipitated using an anti-Nrf2 antibody from the control (middle open bar) and Nrf2 activation (right blue bar) groups were compared. Data are means ± SD and scatterplots with full dataset. (N = 3). * *p* < 0.05 between samples. NS, no significant difference.

**Figure 4 antioxidants-13-00896-f004:**
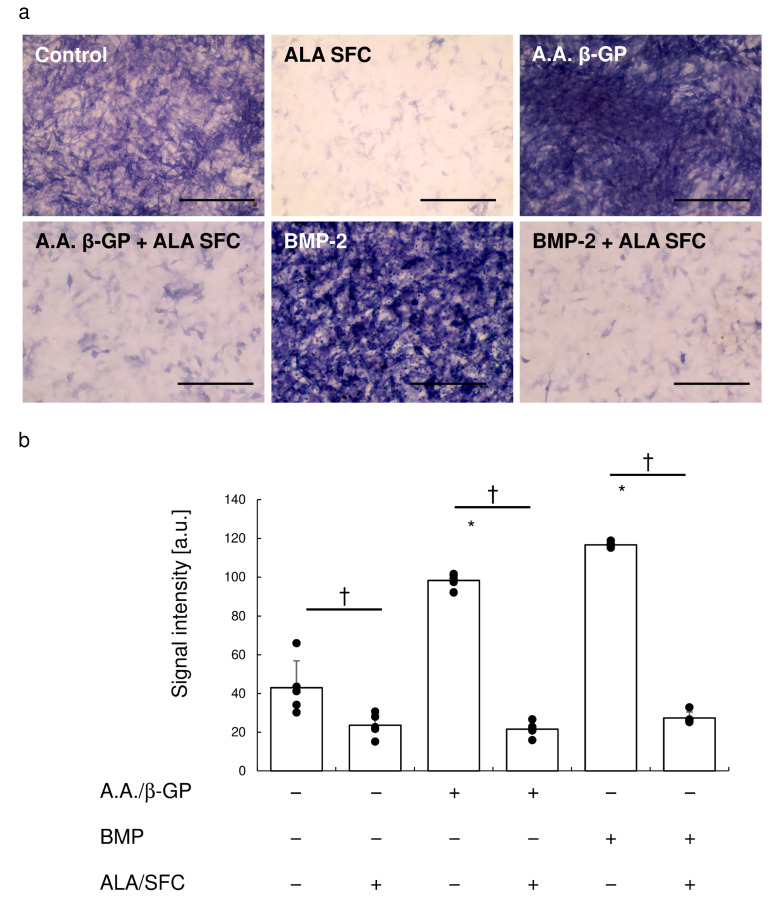
Alkaline phosphatase staining on day 10. (**a**) Representative photographs of cultured cells stained for ALP. Bar, 100 μm. (**b**) Signal intensity measured using ImageJ. Data are means ± SD and scatterplots with full dataset. (N = 5). * *p* < 0.05 vs. control, † *p* < 0.05 between samples.

**Figure 5 antioxidants-13-00896-f005:**
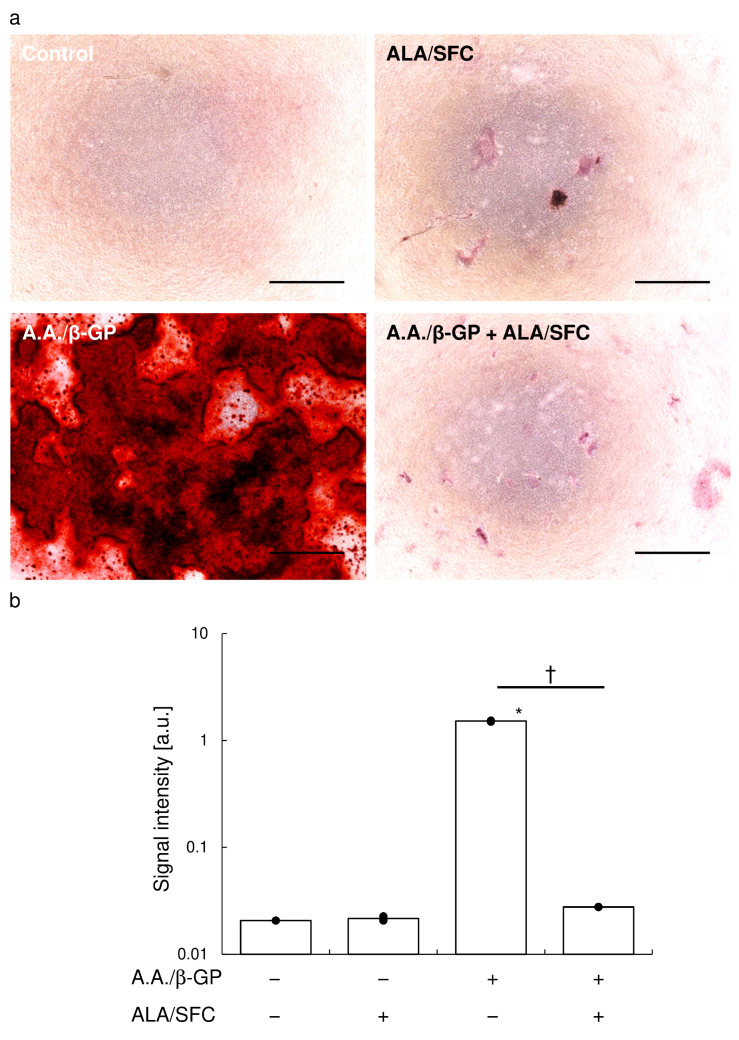
Alizarin red staining on day 28. (**a**) Representative photographs of cultured cells stained with Alizarin red. Bar, 100 μm. (**b**) Signal intensity measured using ImageJ. Data are means ± SD and scatterplots with full dataset. (N = 3). * *p* < 0.05 vs. control, † *p* < 0.05 between samples.

**Figure 6 antioxidants-13-00896-f006:**
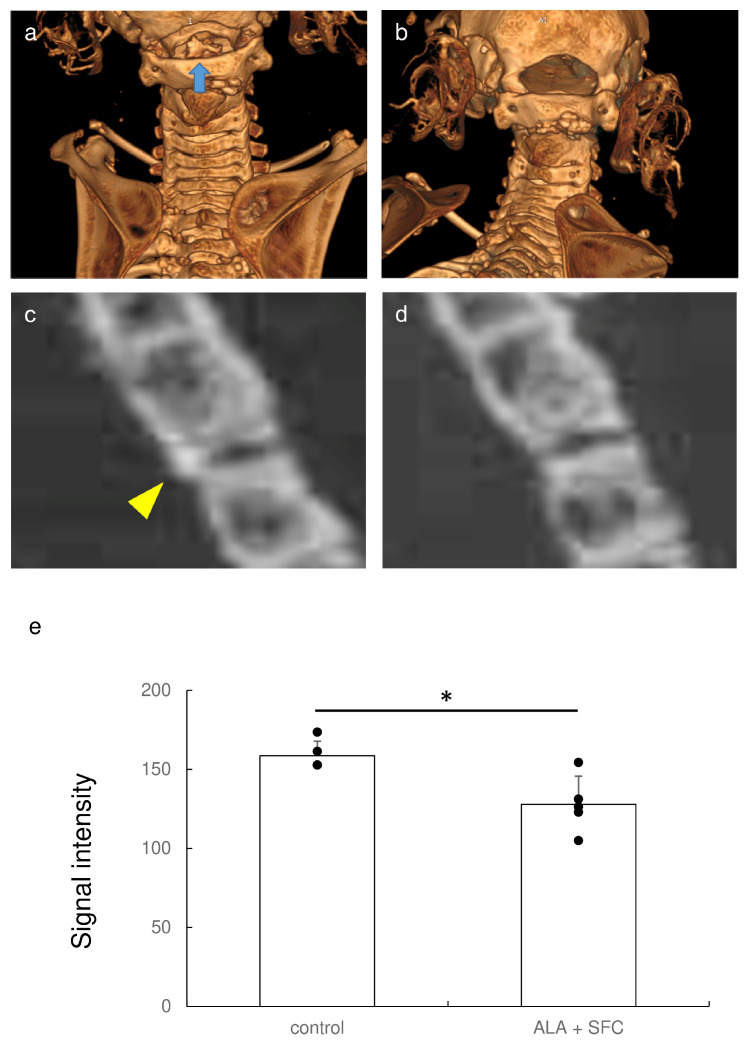
Nrf2 activation in ttw mice attenuated ectopic calcification. Representative 3D-rendered μCT images of control ttw mice (**a**) and ttw mice with Nrf2 activation (**b**). Blue arrow, ectopic calcification. Representative 2D μCT images of control ttw mice (**c**) and ttw mice with Nrf2 activation (**d**). Yellow arrowhead, ectopic calcification. (**e**) Ectopic calcification in ligament tissue between the second and third cervical vertebrae measured using ImageJ (n = 5 each). Data are means ± SD and scatterplots with full dataset. * *p* < 0.05 vs. control.

## Data Availability

The datasets used and/or analyzed during the current study are available from the corresponding author upon reasonable request.

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
