# Peer review of "Activation of Nuclear Factor Erythroid 2-Related Factor 2 Transcriptionally Upregulates Ectonucleotide Pyrophosphatase/Phosphodiesterase 1 Expression and Inhibits Ectopic Calcification in Mice"

_antioxidants, 2024, doi:10.3390/antiox13080896_

Round 1
Reviewer 1 Report
In view the authors demonstrate that Nrf2 activation augmented ENPP1 expression, they referred in introduction
- line 70 that ‘oxidative stress downregulates ENPP1 expression’, meanwhile, during oxidative stress, Nrf2 is activated, and the expression of ENPP1 should be increased.
- This discrepancy should be explained.
Ida Tomomi et al.'s manuscript explores the role of Nrf2 in regulating ENPP1 expression and influencing ectopic calcification. Experiments were conducted using mouse osteoblastic cells and ttw mice. Techniques included real-time RT-PCR, ChIP-qPCR, and immunofluorescence.
Authors demonstrate that Nrf2 activation increased ENPP1 expression, inhibited osteoblastic differentiation, and reduced ectopic calcification.
They concluded that Nrf2 has therapeutic potential to prevent ectopic calcification by activating ENPP1.
However, some discrepancies with the literature need to be explained/commented.
Under oxidative stress, Nrf2 translocation into the nucleus increases, where it heterodimerizes with small MAF proteins. These proteins are required for Nrf2-associated activation of antioxidant response element (ARE)-dependent target genes.
In view the authors demonstrate that Nrf2 activation augmented ENPP1 expression, they referred in introduction
- line 70 that ‘oxidative stress downregulates ENPP1 expression’, meanwhile, during oxidative stress, Nrf2 is activated, and the expression of ENPP1 should be increased.
- This discrepancy should be explained.
- In Figure 2 two types of putative Nrf2 binding sites is highlighted. On the left both are labeled 1, 2, etc. Do they have similar binding affinities to Nrf2 and do they represent two separate putative binding sites called 1
Minor
The ANK gene symbol should be described.
Reviewer 2 Report
This is an interesting paper on inhibition of ectopic calcification by NRF2.
For the most part, the methodology is adequately described. The data appear to be adequately collected and properly analysed. However, the paper would benefit from some revisions.
Better description of in vivo experiments is required.
The paper would benefit from the section limitations.
Some extended discussion is suggested. For example, downstream of which nuclear receptor would be the predicted activation of NRF2.
I suggest to discuss vitamin D, since classical active form can induce ectoipic calcifications. However, novel recently described CYP11A1 derived forms do not, and tyhey also activate NFR2 ( see: Biological Effects of CYP11A1-Derived Vitamin D and Lumisterol Metabolites in the Skin, Journal of Investigative Dermatology, 2024, https://doi.org/10.1016/j.jid.2024.04.022.)
For the most part, the methodology is adequately described. The data appear to be adequately collected and properly analysed. However, the paper would benefit from some revisions.
Better description of in vivo experiments is required.
The paper would benefit from the section limitations.
Some extended discussion is suggested. For example, downstream of which nuclear receptor would be the predicted activation of NRF2.
I suggest to discuss vitamin D, since classical active form can induce ectoipic calcifications. However, novel recently described CYP11A1 derived forms do not, and tyhey also activate NFR2 ( see: Biological Effects of CYP11A1-Derived Vitamin D and Lumisterol Metabolites in the Skin, Journal of Investigative Dermatology, 2024, https://doi.org/10.1016/j.jid.2024.04.022.)
